# Niosomal Drug Delivery Systems for Ocular Disease—Recent Advances and Future Prospects

**DOI:** 10.3390/nano10061191

**Published:** 2020-06-18

**Authors:** Saliha Durak, Monireh Esmaeili Rad, Abuzer Alp Yetisgin, Hande Eda Sutova, Ozlem Kutlu, Sibel Cetinel, Ali Zarrabi

**Affiliations:** 1Nanotechnology Research and Application Center (SUNUM), Sabanci University, Istanbul 34956, Turkey; salihadurak@sabanciuniv.edu (S.D.); esmaeilirad@sabanciuniv.edu (M.E.R.); yalp@sabanciuniv.edu (A.A.Y.); esutova@sabanciuniv.edu (H.E.S.); ozlemkutlu@sabanciuniv.edu (O.K.); 2Faculty of Engineering and Natural Sciences, Molecular Biology, Genetics and Bioengineering Program, Sabanci University, Istanbul 34956, Turkey; 3Faculty of Engineering and Natural Sciences, Materials Science and Nano-Engineering Program, Sabanci University, Istanbul 34956, Turkey; 4Center of Excellence for Functional Surfaces and Interfaces (EFSUN), Faculty of Engineering and Natural Sciences, Sabanci University, Tuzla, Istanbul 34956, Turkey

**Keywords:** niosome, nanocarrier, ocular drug delivery, eye disease

## Abstract

The eye is a complex organ consisting of several protective barriers and particular defense mechanisms. Since this organ is exposed to various infections, genetic disorders, and visual impairments it is essential to provide necessary drugs through the appropriate delivery routes and vehicles. The topical route of administration, as the most commonly used approach, maybe inefficient due to low drug bioavailability. New generation safe, effective, and targeted drug delivery systems based on nanocarriers have the capability to circumvent limitations associated with the complex anatomy of the eye. Nanotechnology, through various nanoparticles like niosomes, liposomes, micelles, dendrimers, and different polymeric vesicles play an active role in ophthalmology and ocular drug delivery systems. Niosomes, which are nano-vesicles composed of non-ionic surfactants, are emerging nanocarriers in drug delivery applications due to their solution/storage stability and cost-effectiveness. Additionally, they are biocompatible, biodegradable, flexible in structure, and suitable for loading both hydrophobic and hydrophilic drugs. These characteristics make niosomes promising nanocarriers in the treatment of ocular diseases. Hereby, we review niosome based drug delivery approaches in ophthalmology starting with different preparation methods of niosomes, drug loading/release mechanisms, characterization techniques of niosome nanocarriers and eventually successful applications in the treatment of ocular disorders.

## 1. Introduction

The human eye consists of several barriers and defense mechanisms, in order to protect this sensory organ from its surroundings. Due to this specific anatomy, it is a great challenge to deliver drugs to different compartments of the eye and treat ocular disorders. Topical eye drops are the most prevalent way of ocular therapy for the anterior part of the eye, which includes the conjunctiva, cornea, sclera, and anterior uvea [1]. However, this route of drug delivery has some limitations. By using eye drops, the sorption time for drugs is approximately two to three minutes. Hence, applied drops are quickly drained from the surface of the eye. Posterior segment ocular illnesses, like diabetic macular edema, age-related macular degeneration, proliferative vitreo retinopathy, glaucoma, and cytomegalovirus infection are the most common causes of visual impairment and blindness. In some cases, the degeneration of the retina and consequent blindness could also be the result of genetic diseases [2]. Since topical ocular drugs face barriers attaining to the posterior segments, intravitreal injections are introduced as an alternative route of drug administration. Nevertheless, it is necessary to design and formulate new drug delivery systems to sustain the drug concentration over extended periods and minimize the number of shots, due to the invasive nature of the injection [3]. Nanotechnology, by providing particular carriers, is advantageous in ophthalmology in providing new ways of transfer and release, extending the drug efficacy and enabling intracellular delivery [4,5]. Additionally, it is possible to protect the drug from degradation both in in vivo circulation and in storage conditions. Nanocarriers like niosomes, liposomes, polymer-based vesicles, and micelles have attracted much attention from researchers around the world due to such advantages. For example, Rodrigues and co-workers succeed in sustained release of a Myocardin-Related Transcription Factor/Serum Response Factor (MRTF/SRF) inhibitor over time and prevention of conjunctival fibrosis in a rabbit model by using a nanocarrier-based formulation. They encapsulated an MRTF/SRF inhibitor, into large unilamellar liposomes to decrease the release rate. Results showed that this liposome-based nanocarrier released the inhibitor over 14 days gradually [6]. Thriving development of nanocarriers loaded with particular drugs, to lower the intraocular pressure in the eye, was reported by Natarajan et al. introducing this structure as a prime sustained-release nanodrug candidate for glaucoma therapy [7]. Vanish et al. similarly investigated the application of polymeric nanomicelles for intraocular and surface ocular disease treatment [8]. These vesicles extend the half-life of drugs in serum by keeping them away from reticule endothelial systems (RESs). In addition, the non-specific adsorption is being decreased significantly by optimizing the components of vesicles or building a multi-functional surface. 

In the current paper, the aim is to introduce niosome nanoparticles as promising nanocarriers for treatment of various ophthalmic diseases. In the following sections, different preparation methods of niosomes and then drug loading/release mechanisms are explained. Important techniques in the characterization of produced niosomes are introduced. Finally, successful applications of niosomal nanoparticles and current problems in their application for ocular delivery are discussed.

## 2. Anatomy of Eye and the Challenges of Drug Delivery to Eye

The eye is composed of three concentric substrates (Figure 1). The outermost part, which is the fibrous tunic, contains the cornea and sclera. The mid covering, which is called the uvea or vascular tunic, includes the iris, ciliary body, and choroid. The internal part of the eye is the retina. The retina uses retinal vessels and choroid blood vessels for oxygenation. The cornea is a clear and dome-shaped surface that covers the foreside of the iris, anterior part of the eye, and the pupil. The cornea is about 0.5 mm thick with an average vertical diameter of 10.5 mm and a horizontal diameter of 11.5 mm. The retina, which is held in a secure and suitable position by the surrounding cornea and sclera, is a light-sensitive nerve lamella, placed at the posterior section of the eye. This part of the eye senses light transforms it to signals, and then transfers them to the brain through the optic nerve. In a broadly accepted categorization, the human eye is divided into two segments: the anterior part of the eye comprises one-sixth of the eye and includes the iris, cornea, aqueous humor, and lens. The posterior portion constitutes the remaining five-sixth of the eye and consists of the retina, vitreous body, back of the sclera, and choroid [9,10,11]. 

There are several barriers between the ocular compartments, between the eye and its host and the external environment. The corneal epithelium with tight junctions and desmosomes and the mucoaqueous tear layer separate and protect the eye from the external environment. Blood–aqueous barrier is formed by the iris blood vessel endothelium, ciliary epithelium, and retinal epithelium. The conjunctive, sclera, and blood–retina barrier separates the eye from the rest of the body. These protecting barriers also prevent the delivery of topical and systemically administered ocular drugs. For instance, eye drops leave the eye surface in 2–3 min resulting in low bioavailability with less than 3%, efficacy [12]. Topical drug transfer to the posterior part of the eye is even harder, and many diseases that attack the posterior section of the eye cannot be treated properly with current methods. Therefore, there is a great interest in drug delivery, particularly in the posterior portion of the eye [13].

The main route of drug delivery to the anterior portion of the eye is the corneal route. Efficiency of this way of drug administration depends on molecular weight, lipophilicity, ionization degree, and charge of the drugs. In order to deliver the larger and hydrophilic molecules, which cannot easily diffuse through the corneal epithelium, the non-corneal and conjunctival route is used despite being less efficient for drug delivery. Thus, the main path for drug delivery to the posterior segment is concluded to be intravenous and intravitreal administrations (Figure 1) [11]. The best example of the disease that can be treated by these routes is exudative age related macular degeneration. It is necessary to transfer anti-vascular endothelial growth factor (anti-VEGF) drugs to the eye, which is widely done by intravitreal injections. However, in clinical practice, there is no optimal compliance of intravitreal injections. In addition, the injection intervals are very long, that lowers the efficacy of the medication. It is not beneficial to deliver small molecule drugs via intravitreal injections as well, because these molecules have short half-lives in the vitreous (a few hours in comparison to one week in the case of antibodies), requiring repeated injections [2]. Due to aforementioned reasons, improved delivery methods are essential.

## 3. Niosomes as New Drug Delivery Systems

Niosomes are nanocarriers formed from the self-assembly of nonionic surfactants in an aqueous medium resulting in closed bilayer structures (Figure 2). At first, niosome production began in the cosmetic industry [14] and then potential applications of niosome in drug delivery were proposed and explored. Niosomes, as a bi/multi-layer nanoparticles are under investigation by many researchers and pharmaceutical companies for numerous applications. Some valuable studies deal with exploring the optimum methods of drug encapsulation and the kinetics of drug/proteins load and release. Furthermore, the capability of niosomes in enhancing the bioavailability and efficacy of drugs attracts the researchers’ attention to apply niosomal drug delivery systems in treatment of severe inflammations and diseases such as different types of cancers [15,16]. For instance, recently, Hadjizadeh et al. developed BSA-loaded niosomes, and showed the ability of this system to release its cargo slowly. It is also indicated that the release profile of niosome-based complexes are controllable by the cholesterol percentage as a formulation parameter [17]. In another study conducted by Farmoudeh et al., methylene blue (MB)-loaded niosomes were prepared. MB has antioxidant effects and is able to decrease reactive oxygen species, which are produced during the inflammatory phase, following skin injury, leading to a delay in wound healing. This study showed that MB-loaded niosome formulations had high encapsulation efficiency and acceptable stability. In addition, according to biochemical and macroscopic studies, the recovery rate of surgical wounds in the niosomal-treated group was higher than that of other groups [18].

Niosomes are classified based on their size or number of bilayers (Figure 2): Small Unilamellar Vesicles (SUV), Large Unilamellar Vesicles (LUV), and Multi-Lamellar Vesicles (MLV) [19]. The particle size of these non-ionic vesicles is in the sub-micron size range. It has been reported that the size of SUVs are in the range of 10–100 nm, while LUVs exhibit 100–3000 nm length, and that of MLVs are greater than 5 µm. In addition, a few giant vesicles, which were greater than 15 µm, have been published [20]. Due to the unique structure of niosome, it is possible to encapsulate both hydrophilic and hydrophobic materials in the particle core and within the bilayer shell, respectively [21,22].

Nonionic surfactants are the main ingredient used in the structure of niosomes. The tails of most surfactants are similar, consisting of a hydrocarbon chain, which can be branched, linear or aromatic. Generally, surfactants are classified according to their hydrophilic head group. Ionic surfactants include cationic surfactants with a positive charge, and anionic ones with a negative charge. Non-ionic surfactants carry no charge groups on their heads. Surfactant choice depends on the Hydrophilic–Lipophilic Balance (HLB), Critical Micelle Concentration (CMC), and Critical Packing Parameter (CPP) quantities. The most frequently used non-ionic surfactants in the preparation of niosomes are listed in Table 1 [23,24].

### 3.1. Preparation of Niosomes

#### 3.1.1. Thin-Film Hydration (TFH) Method

The TFH method was first introduced by Bangham et al. for the preparation of liposomes [25]. TFH method is one of the most widely and simple methods utilized for the preparation of liposomes, which could be used for the preparation of the niosomes, as well. In TFH method, the surfactant and other additives are homogeneously dissolved in an organic solvent such as chloroform, or a mixture of organic solvents, in a round-bottom flask. Then, the solvent is evaporated completely using a rotary vacuum evaporator, and the thin film is obtained at the inner surface of the flask. The thin-film is then re-hydrated with an aqueous medium including water or phosphate buffer saline (PBS), which commonly contain the drug for encapsulation. After the re-hydration is accomplished, MLV niosomes with various diameters are formed [26,27]. Recently, Ramadan et al. utilized TFH method for production of niosomes with various kind of surfactants and different ratios of surfactants to cholesterol. They encapsulated drug, timolol maleate which is used for treatment of glaucoma by lowering intraocular pressure (IOP), with an encapsulation efficiency (EE%) of 98.8% [28]. 

#### 3.1.2. Solvent Injection (SI) Method

In the SI method (shown in Figure 3), the solvents such as diethyl ether, ethanol are used for dissolving the surfactants and other additives [29]. The homogenous solution is then put inside a syringe pump and is injected drop-wise through a needle to an aqueous solution (may contain drug) at a constant temperature, which is higher than the boiling temperature of the organic solvent. The remaining organic solvent is completely evaporated by a rotary vacuum evaporator. During this evaporation process, unilamellar vesicular niosomes are formed with different sizes, and the entrapped aqueous volume is relatively higher than other methods [30,31,32]. Kakkar and Kaur utilized this method for preparation of elastic niosomes composed of span 60 and tween 80 for encapsulation of lipophilic drug, ketoconazole. They used ethanol as solvent for span 60 and ketoconazole, the solution of which being injected into aqueous phase containing tween 80 [33].

#### 3.1.3. Reverse Phase Evaporation (REV) Method

The *Rev.* method was described by Szoka and Papahadjopoulos in 1978, for the preparation of LUV [34]. Two phases, namely organic and aqueous, are prepared beforehand (as shown in Figure 4). The organic phase is made from a mixture of ether and chloroform, containing a solution of surfactants and additives for membrane formation. The aqueous phase is usually water or PBS, in which the drug is dissolved inside. The organic phase is mixed with the aqueous phase, and then the mixture is vigorously shaken or sonicated to obtain an emulsion. Then, the organic phase is slowly evaporated by a rotary vacuum evaporator at a constant temperature, during which LUV niosomes are begun to form. The evaporation process is completed when the hydration of all of the niosomes are completed [35,36,37]. Villate-Beita et al. prepared niosomes with *Rev.* method. The cationic lipid DOTMA and squalene were dissolved inside dichloromethane, and non-ionic surfactants were put into aqueous phase. Then, dichloromethane is added to the aqueous phase and emulsified. After the niosomes are prepared, plasmid DNA is added inside niosome solutions to obtain nioplexes [38].

#### 3.1.4. The Bubble Method

In the bubble method, niosomes are prepared without the help of organic solvents. The surfactants and additives are mixed in an aqueous phase such as PBS, and then the solution is transferred to a three-neck round-bottom flask. The three-neck flask is then positioned inside a water bath for controlling the temperature. The dispersion of surfactants and additives are occurred at 70 °C. At the start, by utilizing high shear homogenizer, the homogenous dispersion is obtained with stirring for 15–30 s, followed by the bubbling with nitrogen gas of solution at 70 °C [39,40,41].

#### 3.1.5. Freeze and Thaw Method

The freeze and thaw method is an improved method for niosome preparation, which is derived from the TFH method. MLV niosomes suspension which was prepared by the TFH method is frozen in liquid nitrogen, and then thawed in a water bath for a number of cycles with short periods of time [42].

#### 3.1.6. Dehydration–Rehydration Vesicles (DRV) Method

The DRV method was first explained by Kirby and Gregoriadis, in which they used SUVs prepared by the TFH method, to form MLVs [43]. Briefly, SUVs, prepared by the TFH method, were separated by centrifugation. Afterward, SUVs were added to the aqueous phase with a drug, and this suspension was freeze-dried overnight. After rehydration of dried product, multilamellar DRVs were generated [44].

#### 3.1.7. Microfluidization Method

The microfluidization method is developed recently for preparation of vesicular particles. In this method, two fluidized streams of organic and aqueous phases are moved forward through to a specific micro-scale channel and are interacted with very high speeds within the interaction chamber. The interface, where the two phases interact with each other and breach the thin liquid film, is arranged in a specific way that energy given to the system remains within the site of niosome production [45,46]. Seleci et al. produced PEGylated niosomes that are encapsulated with topotecan for anti-glioma treatment, with utilizing microfluidic channel. Briefly, they dissolved span 60, cholesterol and PEG in chloroform (organic phase), and topotecan in aqueous phase. Then, organic phase and aqueous phase are given from different inlets and mixed inside the microfluidic channel, and channel is heated to 65 °C. The prepared niosomes collected from outlet. The drug EE% of niosomes were higher than 37.5% with sizes between 100 and 200 nm [47].

#### 3.1.8. Supercritical Carbon Dioxide Fluid (scCO_2_) Method

In recent years, the scCO_2_ method was first demonstrated by Manosroi et al. as a novel niosome preparation method [48]. Briefly, they put surfactant, cholesterol, PBS with glucose and ethanol into a glass view cell, which had two windows and fixed volume. Then, CO_2_ was introduced to the system’s view cell, while the pressure and the temperature are maintained at 200 bar and 60 °C, respectively. Niosomes are obtained after 30 min of magnetic stirring and the pressure is then released. LUV niosomes are obtained by this method with a size range from 100 to 440 nm. The main advantage of the scCO_2_ method is the one-step process that uses no toxic, inflammable, and volatile organic solvent [48,49].

#### 3.1.9. Heating Method (HM)

The HM is developed recently by Mozafari for the preparation of nano-carrier systems [50]. Briefly, the surfactants, cholesterol, and drug are added to the aqueous phase such as PBS, which is shown in the Figure 5. The solution is prepared by stirring and heating the aqueous phase. Then, 3% *v*/*v* polyol such as glycerol is added to the solution. This method does not use any toxic, volatile organic solvent, and is introduced as an easy one-step process [49,51].

#### 3.1.10. Pronisome Method

Proniosome has been utilized as a stable precursor for the production of niosomes for drug delivery applications. Proniosomes are made by coating some kind of water-soluble molecule (carrier) e.g., sorbitol, maltodextrin or mannitol, with non-ionic surfactants (shown in Figure 6). The product of this method is dry, free-flowing formulations of a thin-film of surfactants coated onto the carriers. Proniosome-derived niosomes are then prepared by rehydrating the proniosome powder inside hot water with agitation, thus MLV niosomes are formed [49,52,53]. The method has various advantages including good chemical and physical stability for longstanding storage, ease of transportation, and suitability to scale-up [24]. Zeng et al. produced tacrolimus loaded niosomes by utilizing proniosome method. As an immunosurpressive agent, tacrolimus (FK506) is used for preventing the rejection reactions of corneal grafts. Briefly, surfactant, lipid, cholesterol and drug were mixed with ethanol. Then PBS added inside the mixture at 65 °C for 5 min. The gel of proniosomes was stored until reconstitution of niosomes [54].

The aforementioned methods are among the most common methods for producing niosomes with application in ocular drug delivery formulations. Some in vitro and in vivo studies using these methods are briefly described in Table 2.

### 3.2. Niosome Drug Loading Methods

#### 3.2.1. Passive Loading

In this method, drug is passively encapsulated inside the niosomal vesicles. During niosome formation, the drug is included in the hydration step, in which the hydration is carried out using phosphate buffered saline containing the predetermined concentration of drug. To reduce the size of the formed niosomes, sonication could be applied during the hydration. Hashemi et al. prepared niosomes with the TFH method and for passive loading studies, dorzolamide containing phosphate buffered saline was used to hydrate the thin film. The hydration phase was carried out for 1.5 h at 65 °C and sonication was applied to reduce the size of the formed niosomes. However, with this method, they were not able to obtain high entrapment efficiency for the ocular delivery of dorzolamide [56]. In another study, niosomes were prepared with TFH method for enhanced ocular delivery of lomefloxacin HCl. To hydrate the thin film, drug-containing buffer solution (pH 7.4) was used and after 1 h integration, the dispersion was stored overnight at 4 °C to complete the hydration step. Entrapment efficiencies with passive loading were obtained between 40.69% ± 3.97% and 68.41% ± 0.07% according to different surfactant types and different surfactant:cholesterol ratios [63]. Since it is a simple method, the passive loading technique is commonly used in ocular drug delivery studies with niosomes. However, the percentage of drug that can be loaded with this method is limited.

#### 3.2.2. Active Loading (Remote Loading)

Active loading refers to the encapsulation of the drug molecules into a niosome with the help of a transmembrane gradient. Drug passes through the niosome membrane by diffusion and accumulates inside the niosome by protonation [85,86]. Compared to passive loading methods, the main advantage of this method is its high drug loading efficiency [86].

• Active Loading with Transmembrane pH Gradient Method

The principle of this method is that the basic drug is ionized and precipitated by entering the acidic environment inside the niosome. Because of the pH difference inside (lower pH) and outside (higher pH) of the niosome, the ionized drug cannot go out and is trapped inside of the vesicle [39,87]. Thus, niosomes loaded with pH gradient display elevated entrapment efficiency and drug retention [86,87]. Lakshmi and Bhaskaran exhibited a niosomal formulation for drug loading with pH gradient method. They prepared niosomes by TFH method and used citric acid buffer at pH 4.0 to hydrate the thin film. After freeze and thawing the vesicles several times, drug-containing aqueous phase was included to niosomal suspension and vortexed. Finally, disodium hydrogen phosphate was used to increase the outside pH to 7.0–7.2. With this method, they achieved higher EE% than other methods that they conducted up to 52.9% ± 2.35% [88].

• Active Loading with Transmembrane Ion Gradient Method

This method involves the precipitation of the drug in the presence of a transmembrane ion gradient, such as a sulfate or phosphate. In a study where ion gradient method was used for loading drugs into liposomal vesicles, it has been shown that the drug is trapped by up to 100% in the presence of sulfate, phosphate, citrate, or acetate ions [85]. Uchegbu et al. showed that loading drugs into niosomes could be implemented with higher entrapment and stability using ammonium sulfate gradient [89]. Similarly, in another study ammonium sulfate gradient was used to drug loading into niosomes. Hydration of thin film was carried out with ammonium sulfate for 1 h at 55 °C using sonication to reduce the size of the formed niosomes. After removal of untrapped ammonium sulfate with dialysis against 10% sucrose, the drug solution was added to the niosomal suspension, and stirred for 1 h at 55 °C. In all the niosomes they obtained, the EE% was successfully above 70% [90]. Hashemi et al. examined loading of dorzolamide, a drug used in the treatment of glaucoma, into niosomes for ocular delivery with passive loading and phosphate gradient method. For phosphate gradient method di-ammonium hydrogen phosphate was used for hydration after thin film formation. Subsequently, drug loading was performed for 1 h adding dorzolamide HCl to niosomal suspension. According to their results, maximum encapsulations were 28.3% ± 1.5% and 47.7% ± 1.3% for passive loading and phosphate gradient methods, respectively [56]. In this case, it may be concluded that drug loading with ion gradient is a more efficient method compared to passive loading.

### 3.3. Characterization of Niosomes

Characterization studies are carried out to determine the average size, size distribution, zeta potential, morphology, stability, and in vitro and in vivo drug release characteristics of developed niosomal vesicles.

#### 3.3.1. Average Size, Size Distribution, and Morphology

The average size and size distribution of niosomes could be determined using dynamic light scattering (DLS), electron microscopy methods such as TEM and SEM, and laser diffraction spectroscopy [19]. 

The size of niosomes varies from 20 to 5000 nm depending on the production method, type of the vesicles, type of the surfactant and the surfactant/cholesterol ratio. Nano-sized vesicles called SUV, micron-sized vesicles called LUV and MLV are niosomal vesicle types that take part in various ocular applications [19,24,91]. 

Vesicle size to be used in ocular applications may vary relying on the application site and intended purpose. For topical applications, niosome size should be of sufficient size against removal with tear turnover and drainage. Fitzgerald et al. presented in their research that MLV have slower removal and better precorneal retention time than SUV [92,93]. In the meantime, Barza et al. showed that LUV having the diameter of 600 nm have prolonged clearance time from the vitreous humor than SUV having the diameter of 60 nm for intravitreal application [94]. With increasing size, systemic drainage is restricted and retention time in the eye increases.

Surfactant type and HLB are other parameters affecting the niosome size. HLB is the main factor affecting the size of niosomes that produced from spans and among them Span 20 has the highest HLB value, followed by Span 40 [95]. For example, size of the obtained from Span 60, Span 40, and Span 20 niosomes for ocular delivery of lomefloxacin HCl by the TFH method has increased with the increasing HLB, respectively [63]. Similarly, niosomes for the ocular delivery of naltrexone hydrochloride in treatment of diabetic keratopathy obtained with Span 40 significantly larger than those obtained with Span 60 [23]. On the other hand, size of the niosomes produced from Tween is affected by alkyl chain length of the surfactant. An increase in length of the alkyl chain usually results in an increase in size [80,95]. For instance, in the case where niosomes formulated using Tween 80, Tween 60, and Tween 40 for ocular delivery of lomefloxacin HCl by TFH method resulted in decreasing size, respectively [63]. 

The homogeneity of the size distribution of the particles is defined by the polydispersity index (PDI) [96]. For ocular applications, particles should have narrow size distribution and small polydispersity index values as lower PDI values are better for the uptake of nanoparticles and posterior internalization [97]. 

To verify size and size distribution characterization, morphological characterization of niosomes could be analyzed by atomic force microscopy (AFM) and different electron microscopy techniques. Studies conducted so far show that niosomes are spherical in shape [98,99,100].

#### 3.3.2. Charge of Niosomes

Particle charge is a critical parameter affecting the interaction of niosomes with biological structures and the drug molecules. The fact is that, charged particles repel each other and generate more stable structures [100]. 

Aggarwal et al. suggested encapsulating acetazolamide into niosomes for glaucoma treatment, and their study has indicated that positively charged niosomes have elevated entrapment efficiency than neutral and negatively charged niosomes for all preparation methods that they conducted due to the interaction between the drug and phospholipids [98].

Besides, positively charged particles are more successful candidates to interact with negatively charged cell membranes [100]. Positively charged niosomes have better retention time than neutral or negatively charged niosomes on negatively charged corneal epithelium [92,93].

#### 3.3.3. Entrapment Efficiency

Entrapment efficiency (EE) is a fundamental parameter for characterization that indicates the drug encapsulation capacity of the vesicle. It is calculated (Equation (1)) by determining the amount of initial drug (*W_i_*) in the suspension, amount of unloaded drug (*W_f_*) and hence the amount of loaded drug using analysis methods including spectrophotometry, high performance liquid chromatography (HPLC), and enzyme-linked immunosorbent assay.
(1)$$EE\%=\left(W_{i}-W_{f}\right)/W_{i}\times 100$$

Vesicle size, surfactant type, alkyl chain length, and cholesterol amount are the parameters affecting entrapment efficiency. Alkyl chain length alters membrane permeability, thereby affecting the encapsulation efficiency. Higher entrapment efficiency is obtained with longer alkyl chain surfactants compared to shorter alkyl chain surfactants. For instance, gentamicin sulfate loaded niosomes for topical ocular applications produced by the TFH method from stearyl (C18) chain surfactants have better EE% than niosomes produced with lauryl (C12) chain surfactants [80]. In addition, niosomes obtained with Span 60 (C16) have higher EE% than those obtained with Span 40 (C14) and Span 20 (C12) because Span 60 has the longest alkyl chain among them [101]. Similarly, Guinedi et al. prepared acetazolamide loaded niosomes for glaucoma treatment and found that niosomes prepared using Span 60 have better EE% in both the *Rev.* and TFH methods [36]. They explained this by the fact that Span 60 has longer alkyl chain and a lower HLB value than Span 40. 

It is shown that an increase in the cholesterol amount induces entrapment efficiency at the first stage due to the attachment of cholesterol to the hydrocarbon chains, which leads to the membrane stability. However, further increasing the amount of cholesterol disturbs the bilayer structure and causes a decrease in entrapment efficiency of niosomes [23,36,80]. For example increasing surfactant:cholesterol ratio from 7:6 to 7:7 decreased EE% of acetazolamide loaded niosomes prepared using Span 40 and Span 60 in both the *Rev.* and TFH methods [36]. Correspondingly, in a study for ocular delivery of naltrexone hydrochloride, increasing the surfactant:cholesterol ratio to 1:1 decreased EE% of niosomes containing Span 40, Span 60, and Span 80 [23].

It is important to consider the hydrophilic or lipophilic potential of the drug when evaluating the EE%. For example, acetazolamide loaded MLVs prepared by TFH method had higher EE% than acetazolamide loaded LUVs or SUVs prepared by *Rev.* method. The acetazolamide used in the treatment of glaucoma is a hydrophobic drug and therefore showed better EE% due to the multilayered hydrophobic bilayers of the MLVs [36]. Similarly, niosomal vesicle formulations of lipophilic gatifloxacin for ocular infections prepared by SI method, showed high EE% [79]. Selecting preparation methods such as TFH and SI, which enables MLVs to encapsulate hydrophobic drugs into the bilayer niosomes, will increase the EE% of the hydrophobic drug. Comparably, timolol maleate, a hydrophilic drug used in the treatment of glaucoma, has been found to have higher EE% in LUVs obtained by the *Rev.* method, when compared to other niosomal formulations obtained by TFH and SI methods. LUVs obtained by the *Rev.* method provide a large aqueous interior and high aqueous volume to lipid ratio for hydrophilic timolol maleate, which explains the high EE% [59,61]. Another hydrophilic drug, doxycycline, was entrapped to niosomes by the *Rev.* method and its EE% was found to be higher than the niosomal formulations prepared by the TFH method. The *Rev.* niosomes encapsulated the hydrophilic drug with higher efficiency, as they have larger aqueous internal volume [81]. The ability of LUVs to encapsulate higher amount of hydrophilic drug within the vesicle should be taken into account in determining the preparation method of niosomal formulations for water-soluble drugs. Consequently, choosing the preparation method considering the hydrophilic or lipophilic potential of the drug will provide higher EE%.

#### 3.3.4. Stability of Niosomes

Stability studies are carried out by analyzing the size, PDI, and zeta potential of niosomes that are stored for a certain period of time at 4 and 25 °C. Storage conditions are crucial for stability and niosomes that stored at 4 °C have maintained their size, zeta potential, and stable structure. However, at 25 °C storage condition, zeta potential values decreased from +40 to −20 mV. With a decrease in zeta potential values, aggregation occurred and therefore increase in size was observed at 25 °C storage conditions after 100 days [100,102].

#### 3.3.5. In Vitro Release

Drug release mechanisms from niosomal systems may have diverse release kinetics such as zero order, first order, Higuchi’s model, and Korsmeyer and Peppas model. The zero order model (Equation (2)) represents the cumulative percentage release against time, while first order model (Equation (3)) represents logarithmic percentage release against time. Higuchi’s model (Equation (4)) represents cumulative percentage release against square root of time, and Korsmeyer and Peppas equation (Equation (5)) represents the logarithmic amount of drug released against logarithmic time. Drug release kinetics are characterized by meeting the in vitro release study data with different release kinetics using following equations:

Zero order:(2)$$W_{t}=W_{0}-K_{0}t,$$
where *W_t_* is the amount of drug released in time *t*, *W*_0_ is the initial amount of drug in solution, and *K*_0_ is the zero order constant;

First order:(3)$$\ln W_{t}=\ln W_{0}-K_{1}t,$$
where *W_t_* is the amount of drug released in time *t*, *W*_0_ is the initial amount of drug in solution, and *K*_1_ is the first order constant;

Higuchi’s model:(4)$$W_{t}=K_{h}\sqrt{t},$$
where *W_t_* is the amount of drug released in time *t*, and *K_h_* is the Higuchi release constant;

Korsmeyer and Peppas model:(5)$$M_{t}/M_{\infty }=K_{kp}t^{n},$$
where *M_t_/M_∞_* is a fraction of drug released in time *t*, *K_kp_* is the Korsmeyer and Peppas release constant, *t^n^* is the release exponent, and *n* indicates drug release mechanism.

Drug release kinetics is also related to the application site, application route, solubility, and dosage of the drug. Zero order release provides a stable concentration in the release site. For first order release kinetics, the rate of release decreases logarithmically over time [103,104,105]. On the other hand, Higuchi’s model [106] indicates that drug release rate slows down as the distance traveled by the drug increases. In this case, drugs close to the vesicle surface are released faster, while those near the core are released more slowly and the system functions as a reservoir for continuous drug release [107,108].

Release kinetic is a critical parameter for ocular applications because it determines drug retention time and therapeutic effect on the ocular administration surface. In a study where the niosomal system was prepared by TFH method and used for the ocular delivery of atenolol, it was stated that the niosomal system had Higuchi’s model release kinetics and release from the niosomal dispersion was continued for 8 h without reaching plateau [57]. In the studies with niosomal formulations of lomefloxacin HCl prepared by TFH method for ocular bacterial conjunctivitis treatment, niosomes showed Higuchi’s model release kinetics [63,78]. Abdelkader et al. proposed niosomal systems for the ocular application of naltrexone hydrochloride and obtained mostly erosion controlled a zero order release kinetics after 12 h in their study [42]. Similarly, another ocular study with niosomes prepared by film hydration method indicated that the niosomal system showed zero order release kinetics and 22% drug release in 8 h [58]. However, more than one release kinetics can prevail in a system [104]. 

In vitro release kinetics can be determined by dialysis membrane or Franz diffusion cell methods. In the dialysis membrane method (Figure 7a), the dialysis membrane containing the drug loaded niosome suspension is incubated in a buffer solution and the released drug amount is determined with a spectrophotometric method or chromatographic method by withdrawing samples from the buffer solution at certain intervals [57,82,83]. The Franz diffusion cell method (Figure 7b) includes a donor, a receiver and a cellophane membrane or dialysis membrane between them. Drug molecules diffuse with passive diffusion from donor unit to buffer solution in the receiver unit [42,109]. Besides, Li et al. and Zubairu et al. carried out a modified Franz diffusion cell method using excised cornea for ocular drug release studies with niosomes [76,79]. In drug release studies, the buffer solution could be prepared at pH 7.4 to mimic tear fluid and at 35 ± 0.5 °C to mimic the ocular surface temperature [42,57].

Niosomal formulations prolonged drug release successfully in several studies in comparison with free drug formulations. Timolol maleate loaded niosomes developed for the treatment of glaucoma showed a prolonged release profile and entire drug was released from the vesicles in about 24 h, while its free form was completely released in 4 h [28]. Free drug solution of brimonidine tartrate, another drug used for glaucoma treatment, shown to release 75% of the drug within 2 h, whereas niosomal formulation released 22% of the drug in 8 h [58]. The free solution of doxycycline, used as an antibiotic in ocular applications, was released almost completely within the first hour, but niosomal formulations showed sustained release for up to 20 h [81]. Similarly, encapsulation of gentamicin sulfate into niosomes showed sustained release and it was released much slowly than its free drug solution [80].

Additionally, the niosomal vesicle size and structure affect the release profile of the encapsulated drug. For instance, the release of acetazolamide was slower from MLVs compared to LUVs or SUVs. MLVs have several multilayered hydrophobic bilayers resulted in prolonged release of the hydrophobic drug entrapped in these bilayers [36].

## 4. Niosome Based Treatment Strategies in Ocular Diseases

### 4.1. Glaucoma

Glaucoma is considered as a neurodegenerative disease and often requires a lifetime treatment [55,110,111]. Glaucoma is an optic neuropathy that is characterized with retinal ganglion cell (RGC) death by apoptosis, resulting in degeneration of nerve axons and visual field defects [110,112,113]. There are different risk factors and various types of glaucoma. The most prevalent factor associated with Primary Open Angle Glaucoma prognosis is the increased intraocular pressure (IOP) [110,113,114]. IOP rises because of the accumulation of aqueous humor in the anterior chamber, which might be due to the overproduction of fluid or the blockage of drainage system [55]. The elevated IOP causes an imbalance in the retinal blood flow, which leads to the degeneration of the optic nerve [55]. The mainstream treatment method for glaucoma is to reduce the intraocular pressure by various treatment options including drugs, laser treatment, and surgery [114,115]. 

Niosomes are investigated as drug delivery vehicles for glaucoma treatment. There are several advantages of the niosomes for drug delivery as topical eye drops such as the less frequent administration, extended IOP-lowering activity, higher corneal permeation, and low ocular toxicity [91]. Aggarwal and Kaur prepared a timolol maleate (a water-soluble drug used for treatment of open angle glaucoma) encapsulated chitosan or carbopol-coated niosome formulations by REV, ether injection, and film hydration methods. Niosomes prepared by using *Rev.* method showed the highest ratio of entrapment efficacy compared to others due to the preparation of LUVs that could encapsulate a water-soluble drug in large quantities. Chitosan-coated niosome formulations exhibited prolonged control over IOP, compared to carbopol-coated niosomal formulations [61]. In another study, Kaur et al., evaluated the ocular absorption kinetics of timolol maleate loaded into a bioadhesive niosomal delivery system. The results indicated that EE% of niosomes was 24.3%, and niosomes extended drug release more than 10 h and IOP-lowering activity up to 8 h with a peak at 3 h [59]. Guinedi and his colleagues prepared formulation of acetazolamide (ACZ), a glaucoma medicine, as multilamellar niosomal structures by the *Rev.* and TFH methods for in vitro studies. Span 40 or Span 60 surfactants are combined with cholesterol at 7:4, 7:6, and 7:7 molar ratios. The results showed that MLV niosomes of Span 60/Cholesterol with a ratio of 7:6 gave the highest entrapment efficiency, and histological examination made on corneal tissues 40 days after niosomal formulation administration showed marginal reversible irritation in the substantia propria of the eye and no major changes in tissues [36]. Later on, Kaur et al. published another study which was focusing on development of vesicular system for effective ocular delivery of ACZ for the treatment of glaucoma. In this study, the niosomes were able to come into complete contact with corneal and conjunctival surfaces, thus able to enhance the corneal absorption of ACZ. The surfactant, span 60, was considered as the factor that may temporarily increase the permeability of the corneal epithelium. Furthermore, the niosomes were coated with the bioadhesive polymer, Carbopol 934P, which was prolonged the contact time of the formulation in the eyes. As a result, carbopol-coated niosomes inhibit an EE% of 43.75%, and showed longer sustained release and better IOP-lowering activity than the marketed free drug solution [60]. In 2010, Prabhu et al., prepared and evaluated the nanovesicles of brimonidine titrate as an ocular delivery system. They used liposomes and niosomes (with Span 60 as surfactant and cholesterol as additive) for preparation of nanovesicles of acceptable shape and size by using TFH method, and evaluated the morphology, EE%, in vitro drug release and in vivo IOP-lowering activity. The results showed that EE% of the niosomes were higher than 32%, and niosomes showed IOP-lowering activity 3 h longer than commercially available drug formulation [58]. 

Recently, Hashim et al. demonstrated the potential use of niosomal in situ gel to be used as an ocular delivery system for atenolol, a β1 adrenoceptor blocker, for the treatment of glaucoma through topical route. TFH method was used to prepare niosomes using Span 60 and cholesterol at different molar ratios. Maximum entrapment efficiency (EE%) obtained from formulated niosomes was 80.7%. When inside the in-situ gel, niosomes further prolonged drug release and IOP-lowering activity more than 8 h, indicating that niosomal in situ gel formulation exhibit the most considerable prolonged decrease in IOP. The results showed that niosomal in situ gel formulation exhibited in vitro release of the drug in a sustained manner, compared to free drug solution and naive polymeric in situ gel. Therefore, this formulation could be a promising approach as delivery system for atenolol in the glaucoma treatment [57]. 

### 4.2. Conjunctivitis

Conjunctivitis is defined as the inflammation or infection of the conjunctiva, which is the translucent mucous membrane located in the sclera [116]. There are different types of conjunctivitis such as viral, bacterial, and allergic conjunctivitis that may occur in acute or chronic forms [117]. Bacterial, viral, fungal, parasitic, and chlamydial conjunctivitis is defined as infectious conjunctivitis [117]. On the other hand, main reasons that cause noninfectious conjunctivitis are allergens, toxicities, and irritants [118]. 

The mainstream treatment for conjunctivitis is the topical administration of antibiotics [64], antivirals (aciclovir, trifluridine, and valaciclovir [119]) and antifungals (polyenes, azoles, imidazoles, triazoles, pyrimidines, and echinocandins) [120]. 

In order to increase the efficacy of these drugs, nanotechnological formulations are being investigated for a while now. Abdelkader et al., aimed to prepare and evaluate Span 60-based niosomes for ocular delivery of naltrexone (NTX). Charged lipids including dicetyl phosphate (DCP) and stearyl amine (STA) were examined together with the surfactant as bilayer membrane additives and four different methods were used for their preparation. The prepared niosomal formulations are shown to have non-irritant properties when applied onto the surface of a 10-day-old hen’s chorioallantoic membrane. The controlled release of NTX from niosomal formulation enhanced its corneal permeability [42]. In a subsequent study, by using the *Rev.* method, niosome formulations of NTX were developed by using different surfactants including Span 60, Solulan C24, sodium cholate, and additives including cholesterol and dicetyl phosphate (DCP). The formulations showed no sign of irritation in vivo and were well tolerated in the cornea [84]. Li et al. produced a proniosome that was developed by utilizing surfactants (poloxamer 188 and lecithin) and cholesterol as stabilizer. These niosome formulations were developed for topical delivery of tacrolimus (FK506) and the EE% of drug was 95.32%. in vivo studies showed no signs of irritation and a good biocompatibility. Additionally, there was an enhancement in the precorneal permeation and drug retention [76]. Khalil et al. studied on the development and evaluation of niosomal formulations that were loaded with Lomefloxacin HCl (LXN) to manage ocular bacterial conjunctivitis. They were utilized design of experiment (DoE) approach for obtaining optimum niosomal formulation between different combinations of surfactants including Tween 40, 60, and 80, and Span 20, 60, and 80 and cholesterol. The results showed that EE% of the tween niosomes (>41%) and drug release rate were higher than span niosomes (>40%). The optimum niosomal formulation was healed the infected eyes better than marketed drug solution [63]. In a later study, Abdelbary et al., they focused on improving the formulation of LXN in niosomal formulations of Span 20, 60, and 80 and they investigated the in vivo performance of optimum formulation in rabbits’ eyes. A 35-fold increase was observed in the antibacterial activity of LXN inside niosomes when compared to free drug and prolonged the drug release up to 8 h [78]. In another study, Yasin et al., compared chloramphenicol loaded niosomes and chloramphenicol eye drops in vitro in albino rabbits. It was observed that EE% of niosomal formulations were >83%, and drug release was prolonged up to 10 h. Niosomes cured disease with fewer administrations than marketed eye drop. Niosomes were found ultimate ophthalmic drug carriers capable to release drug in sustained and determined pattern [64]. 

Niosomes formulated with coating are also investigated for increased adhesive and penetration characteristics. Recently, Zeng et al. developed hyaluronic acid (HA)-coated niosomes for the topical delivery of tacrolimus. HA-coated niosomes enhanced the mucoadhesion and prolonged the residence time considerably compared to non-coated niosomes. Additionally, enhanced corneal permeability and decreased drug clearance rate in aqueous humor were observed [54]. Further studies also investigated the effect of HA-coating on niosomes for pharmacokinetics and bioavailability of loaded drugs. Liu et al. aimed to design HA-coated spanlastics (CHASVs) with the immunosuppressive peptide cyclosporine A (CsA), which is used for the treatment of various eye diseases such as corneal graft rejection, uveitis, and dry eye. As a result, CHASVs showed a considerable therapeutic effect and better tear production in dry eye. HA-coated elastic niosomes had EE% of >92%, and HA-coated formulations showed were higher corneal permeability than drug emulsion and Span60 niosomes [75]. In another study, chitosan-coated Span 60 niosomes loaded with gatifloxacin, an ophthalmic solution used for bacterial eye infections, were formulated by solvent injection method. The transition of the drug through cornea was considerably increased with chitosan-coated compared to uncoated niosomes and no toxicity was observed [79].

Allam et al. aimed to prolong the pharmacological action of vancomycin via incorporating drug-loaded niosomes into pH-sensitive in-situ forming gel. The formulation exhibited an unexpected physicochemical stability. In addition, higher in vitro and in vivo antibacterial efficacy was observed compared to the vancomycin hydrochloride-free drug solution and EE% of niosomes was >46%. Moreover, niosomes prolonged drug release more than 24 h [77]. In 2017, Salem et al., formulated lomefloxacin HCl (LF) in niosome form and investigated the in vivo performance in rabbits’ eyes. Span 20, 60, and 80 were mixed together and as additive, cholesterol was used. The results showed a 35-fold increase in the antibacterial activity of LF niosomes in comparison of free drug, drug EE% of niosomes were >78.1%, and niosomes prolonged drug release up to 8 h [63,78]. In a recent study, Akhtar et al. explored utilization of niosomes loaded with azithromycin-β-CD complex. The formulation was optimized through DoE approach and incorporated into the temperature sensitive in situ gel and evaluated. The optimized formulation exhibited superior in vitro drug release profile compared to marketed eye drop. The EE% of prepared niosomes (>30%) and niosomal in situ gel formulations (>63%) were high, and niosomes prolonged drug release up to 12 h with increased corneal permeation. The results indicate that temperature-sensitive niosomal in-situ ocular gel showed increased residence time and provide localized drug delivery effective for the treatment of bacterial conjunctivitis [62].

### 4.3. Retinal Diseases

Inherited Retinal Diseases (IRD) are heterogeneous disorders, resulting in retinal degeneration. Mutations occurred in genes specific for the inner retinal layer, may result in IRD. However, in most cases, the reason is mutated genes expressed in the photoreceptor or retinal pigment epithelium (RPE) cells. For instance, 30 genes have been associated with retinitis pigmentosa, which is the most common form of IRD [121]. 

Retinal gene delivery with niosome carriers for non-viral vectors is investigated. Puras et al. formulated a novel cationic niosome to be used in gene delivery to retina. They used Tween 80 as the surfactant, cationic lipid 2,3-di(tetradecycloxy)propan-1-amine, and squalene as additives, and pCMS-EGFP plasmid for formulation with O/W (oil in water) emulsification method. In vitro and in vivo studies to evaluate transfection efficiency and internalization mechanism are completed in HEK-293 (Human Embryonic Kidney 293) and ARPE-19 (Adult Retinal Pigment Epithelial 19) cell lines, and in rat eyes following subretinal injections, respectively. The results indicated that cell viability was higher for nioplexes, but transfection efficiency was lower compared to Lipofectamine 2000. In vivo transfection efficiencies were depended on route of administration, whereas subretinal injections transfected RPE layer and intravitreal injection transfected inner layers of retina [74]. Mashal et al. aimed to enhance the retinal gene delivery by lycopene incorporation into cationic niosomes based on DOTMA and polysorbate 60 non-ionic surfactant. ARPE-19 cells were used in in vitro experiments after complexion of niosomes with pCMS-EGFP plasmid. Lycopene increased transfection efficiency of nioplexes, but the efficiency was lower than lipofectamine 2000. Nioplexes were transfected into the cells in inner layers of retina. Subretinal and intravitreal administrations to the rat retina indicated that nioplexes had ability to transfect efficiently the outer segments of the retina [73]. In another study, Mashal et al., tested chloroquine diphosphate in cationic niosome formulation made of poloxamer 188, polysorbate 80, and 2,3-di (tetradecyloxy) propan-1-amine (hydrochloride salt) cationic lipid, for transfection into rat retina. Chloroquine is an endosomal disrupting molecule and lysosomotropic agent that can cross the blood retinal barrier when formulated in diphosphate form either with DPP-80-CQ or DPP80. Nioplexes formulated with and without chloroquine were tested on ARPE-19 cells and higher transfection efficiencies were observed with DPP80-CQ nioplexes compared to DPP80 complexes [72]. Recently, a study was done to understand the effects of non-ionic surfactant of niosome components on their transfection efficiency in rat retina. Three niosome formulations that only have difference in the non-ionic tensioactives were prepared. Niosomes contained: cationic lipid 1,2-di-O-octadecenyl-3-trimethylammonium propane (DOTMA), helper lipid squalene and polysorbate 20, polysorbate 80, or polysorbate 85. As a result, nioplexes composed of polysorbate 20 niosomes were the most efficient transfecting retinal cells (ARPE-19) in vitro. Moreover, as they were administrated in subretinal and intravitreal ways, also high levels of transgene expression in rat retinas was observed. Therefore, this formulation generates as a potential non-viral candidate to transfer specific therapeutic genes with a high efficiency into the eye [38].

### 4.4. Keratitis

Keratitis occurs in patients as inflammation in the corneal which can be generated by infections in the presence of pathogens such as bacteria, fungi, and viruses, and constitutes one of the major causes of blindness worldwide [5,122]. Mostly known pre-disposing factors that induce the keratitis are ocular trauma, contact lens usage, and using topical steroids [123,124]. Clinical presentations may vary depending on the infectious agent that causes keratitis, however patients usually show similar symptoms in the eye such as redness, pain, blurry vision, and tearing [125].

In 2017, Paradkar et al. developed a formulation and evaluation of natamycin loaded niosomal in situ gel for ophthalmic drug delivery for the treatment of fungal keratitis. It was observed that EE% of niosome formulations were high (>65%), and in situ gel niosome formulation extended drug release up to 24 h with increased corneal permeation [67]. In 2019, Amit Verma et al. developed natamycin (Nat) loaded trimethyl chitosan (TMC) coated mucoadhesive cationic niosomes (Muc-cat-Nios) for prolonged and effective delivery to the eye. They characterized uncoated and TMC-coated niosomes for mucoadhesive properties and observed an increase. Both uncoated and TMC-coated niosomes extended drug release up to 12 h, while TMC coated niosomes showed higher mucoadhesion and corneal permeability than uncoated niosomes [65]. In the same year, H. Elmosatasem performed a study to improve and prolong the ocular availability of Fluconazole (FL) that is used for the treatment of fungal keratitis. FL-loaded niosomal vesicles were prepared with Span 60, and polymeric nanoparticles were prepared using cationic Eudragit RS100 and Eudragit RL100. FL-HP-β-CD complex was encapsulated in either selected Eudragit nanoparticles (FL-CD-ERS1) or niosomal vesicles. The EE% of the chitosan-coated niosomes was 61.7%, which was lower than Eudragit nanoparticles. The niosomes were further coated with cationic chitosan and these chitosan-coated niosomes prolonged drug release up to 24 h [66]. In another study, novel surfactant-based elastic vesicular system was developed and evaluated for ocular delivery of fluconazole. The highest EE% was 65.73% in prepared formulations, and corneal permeation of niosomes was higher than marketed drug. It was indicated that the developed system enables an effective and safe formulation of fluconazole [71]. 

In another study, voriconazole-loaded niosomes were incorporated into in situ gelling ocular inserts for the treatment of keratitis. Niosomes were prepared from various combinations of surfactants such as Span 60 and Span 40 with pluronic L64 and pluronic F127. The results showed that EE% of the prepared formulations was higher than 49%. and, niosomes and niosomal in situ gel formulations prolonged drug release up to 8 h [68]. Very recently, Gugleva et al. prepared and evaluated niosomes for ocular delivery of doxycycline which is used for the treatment of ocular surface diseases such as rosacea and dry eye. Various niosomes were prepared using Span 20, Span 60, Span 80, and Tween 60 surfactants and their combinations with cholesterol in different molar ratios. Prepared niosomes were well tolerated by the eye. Span60 niosomes exhibit the highest EE% (>50%), and niosomes extended release of drug more than 20 h [81].

## 5. Conclusions and Future Perspectives

Medications applied through topical ocular route are important for lowering patient compliances. However, due to the various ocular barriers, the treatment efficiency of a given drug dose is reduced. Therefore, most of the commercialized topically-applied ocular drugs have drawbacks such as high dosage or high administration frequency. In recent years, there is a great effort for the development of novel drug delivery systems including niosomes for the treatment of ophthalmologic diseases. The extensive effort is continuing especially for developing eye drops. Herein, we reviewed a number of in vitro and in vivo studies of niosomes for a variety of ophthalmic diseases therapy.

Niosomes are versatile, easy to prepare, biocompatible nanocarriers with sizes change from tens of nanometers to few micrometers. The niosomes are utilized for drug and gene delivery to both anterior and posterior segments of the eye with increased corneal permeation, higher ocular bioavailability of the drug, and prolonged drug release. Additionally, the advantages of niosomes are further improved with the methods including coating niosomes with some kind of mucoadhesive polymer such as chitosan, carbopol, or HA or incorporating niosomes into in-situ gels [54,59,60,61,65]. 

Compared to numerous nanostructures utilized for ocular drug delivery formulations, niosomes exhibit superior chemical stability and spreading on the surface of the eye and thereby have high bioavailability [126]. Niosomes are also more capable of penetrating than the others. Even though, niosomes and liposomes are structurally similar, the materials used for preparation of niosomes are more stable [40]. For example, the susceptibility of phospholipids of liposomes to oxidative degradation is one of the considerable problems while using liposomes as vesicular system. Liposomes need to be stored and handled in an inert atmosphere such as nitrogen. In addition, purification of them is costly, whereas, niosomes and their production are cheaper and simpler [127]. These advantages of niosomes such as cheaper building blocks and higher stability are pulling researchers to work on niosomes.

However, niosomes should be investigated in further detail before they could take place in applications in the ocular field. Although they are more stable than liposomes in drug delivery applications, there are still problems to be solved related to niosomes. In fact niosomes cannot maintain their stability at room temperature and the occurrence of vesicle aggregation over time indicates that appropriate storage conditions are needed [100,102,128]. Failure to maintain stability may result in drug leakage and a decrease in the level of encapsulated drug. Until now, most of the niosomal formulations are made for increased bioavailability of the drug in ocular tissues. However, for the treatment of ophthalmic diseases, drug molecules need to be delivered to the specific part of the eye, such as retina in the case of macular degeneration. Therefore, targeting strategy needs to be integrated into the niosomal formulations. There are also ongoing researches for elastic niosomes, which are more efficient than rigid ones for drug delivery to the posterior segment of the eye. With further in vivo studies, it is necessary to develop suitable niosomal dispersions for ocular topical and intravitreal administration routes. Since niosomes are composed of surfactants, determination of appropriate surfactant and surfactant:cholesterol ratio are the subjects that should be improved in order to achieve low toxicity, high stability, and high efficiency. Emphasis on biocompatible, biodegradable and lower cost surfactants will enable the development of successful niosome formulations. Furthermore, approaches to improve the therapeutic potential and bioavailability of niosomes should focus on improving the ability to pass through the ocular barriers and prolonging retention time. However, it should also be examined whether the increased retention time causes systemic side effects. Despite the fact that, in vivo ocular toxicity tests showed no irritation, inflammation, or redness for niosomal formulations [79,84,128], further toxicity assessment is needed for forthcoming preclinical or clinical studies. Hence, these strategies might be the future of the niosome researches in ophthalmology. Improvements in stability and therapeutic efficiency will pave the way for large-scale production of niosome drug formulations for clinical use.

## Figures and Tables

**Figure 1 nanomaterials-10-01191-f001:**
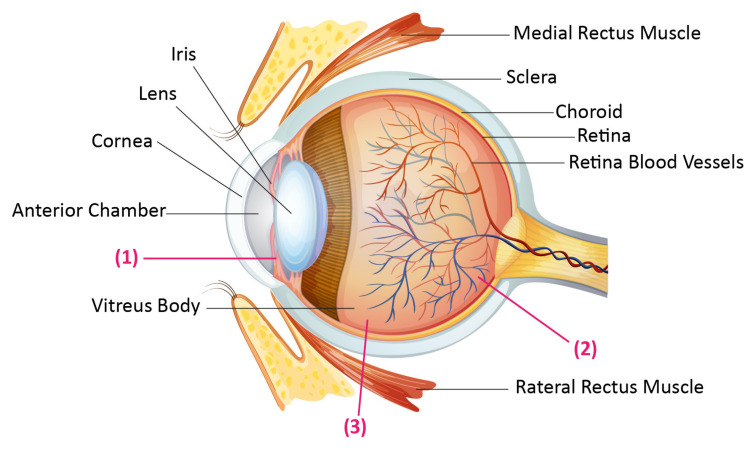
Human eye anatomy, and the drug administration routes. (1) The cornea as a main route of drug delivery to the anterior section. (2) The retinal capillary endothelium and Retinal pigment epithelium as the main barriers for systemically administered drugs. (3) An invasive strategy to gain the vitreous called intravitreal injection.

**Figure 2 nanomaterials-10-01191-f002:**
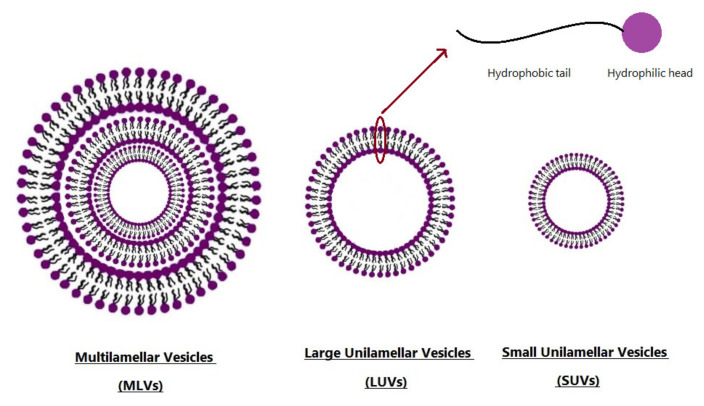
Typical vesicle size of niosomes.

**Figure 3 nanomaterials-10-01191-f003:**
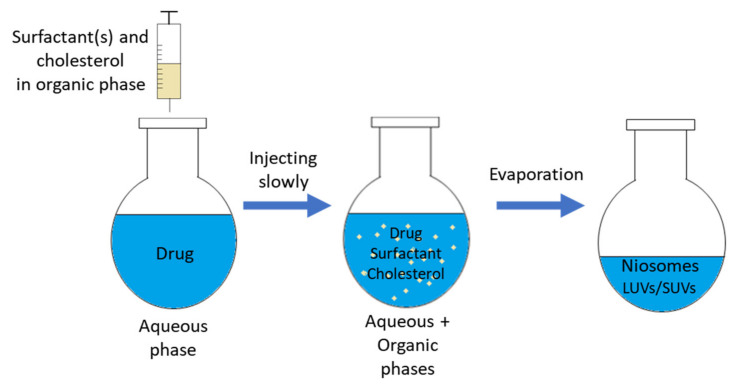
Schematic representation of the solvent injection method.

**Figure 4 nanomaterials-10-01191-f004:**
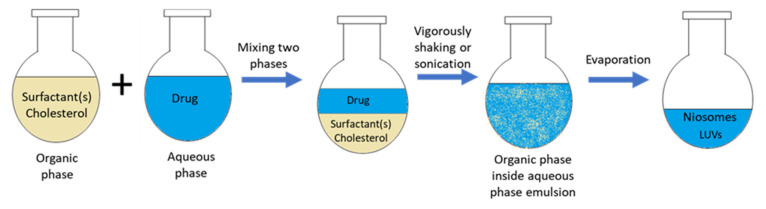
Schematic representation of reverse-phase evaporation method.

**Figure 5 nanomaterials-10-01191-f005:**
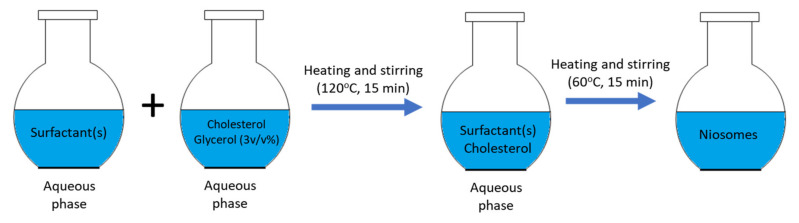
Schematic representation of heating method.

**Figure 6 nanomaterials-10-01191-f006:**
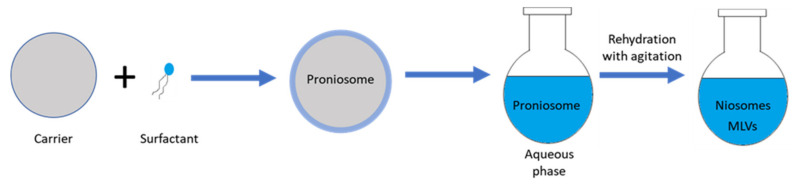
Schematic representation of proniosome method.

**Figure 7 nanomaterials-10-01191-f007:**
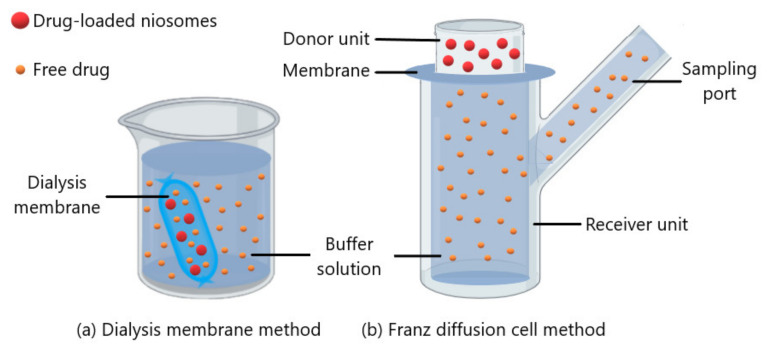
Schematic representation of (**a**) dialysis membrane method and (**b**) Franz diffusion cell method.

**Table 1 nanomaterials-10-01191-t001:** Surfactants commonly used for niosome preparation.

Polysorbates (Tweens)		Sorbitans (Spans) *	
**Tween 20** CMC~0.06 mM	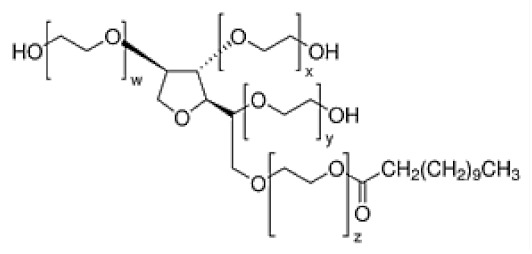	**Span 20**	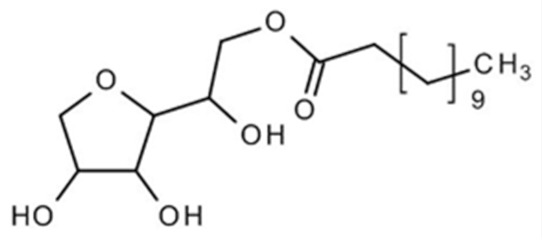
**Tween 40** CMC~0.027 mM	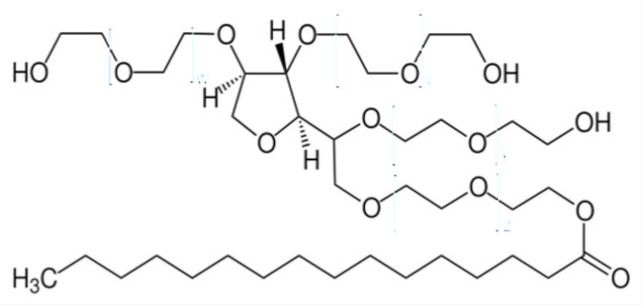	**Span 40**	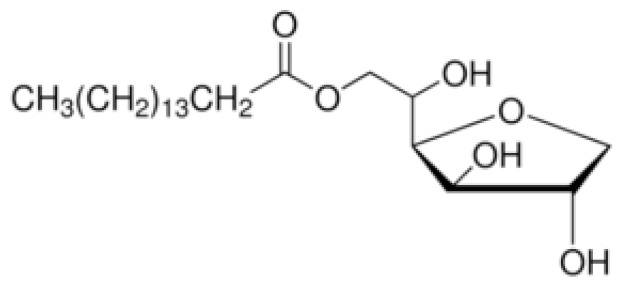
**Tween 60** CMC~0.026 mM	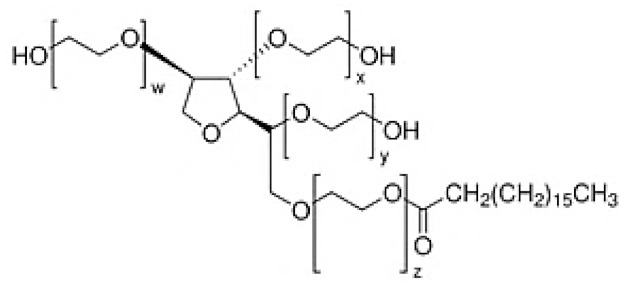	**Span 60**	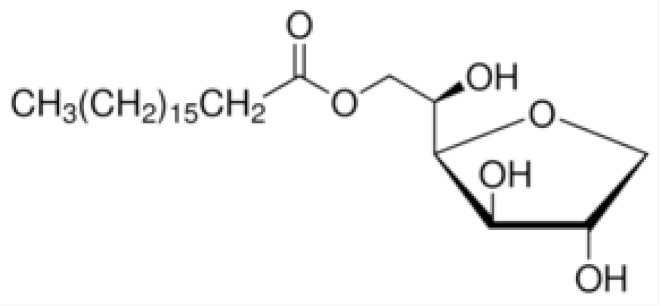
**Tween 80** CMC~0.012 mM	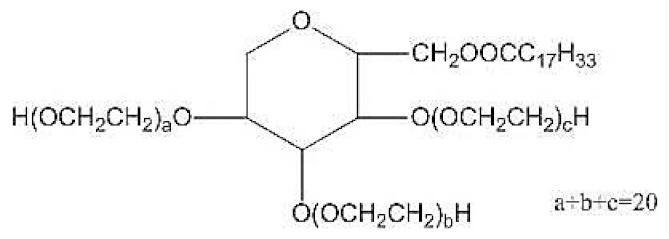	**Span 80**	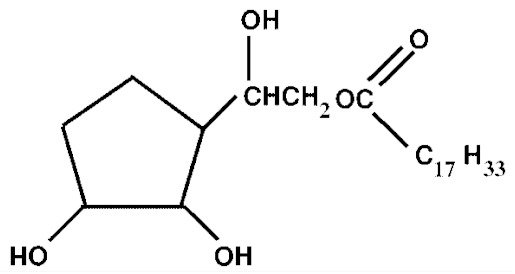
**Tween 65** CMC~0.00018 mM	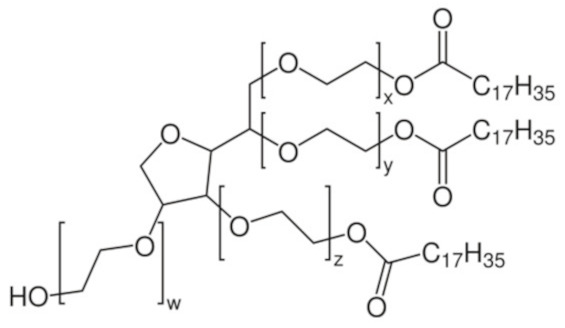	**Span 65**	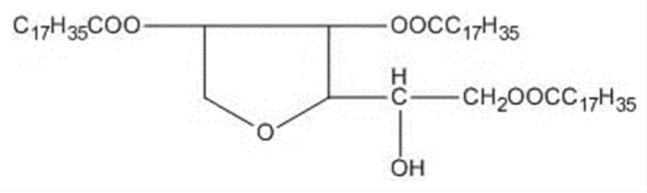
**Tween 85** CMC~0.00029 mM	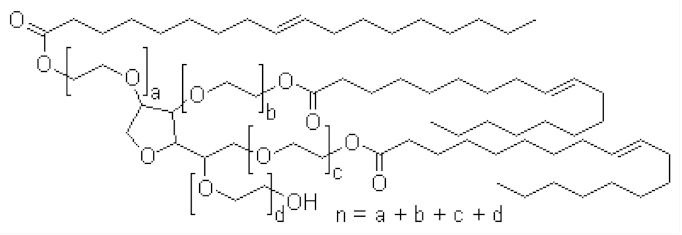	**Span 85**	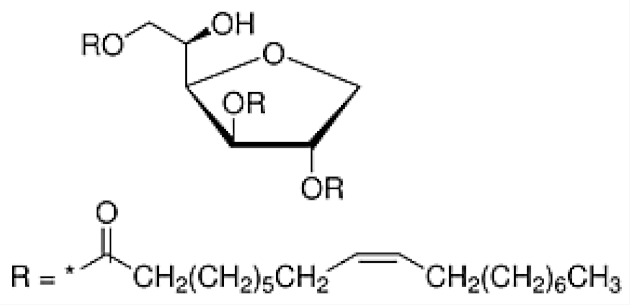

* Due to ethoxylation, the Tweens are water soluble opposite to the oil soluble spans. Therefore, Critical Micelle Concentration (CMC) is not an applicable parameter for spans.

**Table 2 nanomaterials-10-01191-t002:** In vivo and in vitro studies of niosomes in ocular drug delivery.

Disease	Surfactant(s)	Additive(s)	Drug	Production Method(s)	Results	Year, Ref.
Glaucoma	Span 20, 40, 60 Tween 20, 40	Cholesterol	Timolol maleate	TFH	Highest EE% found in niosomes prepared with span60 and Tween 40 (>90%), and niosomes prolonged drug release and IOP-lowering activity up to 24 h.	2020, [28]
	Span 60 Brij 52	Cholesterol Soybean α-lecithin	Brimonidine tartrate	Proniosome	Highest EE% obtained be span60 formulations (>64%) as compared to brij52 niosomes. Niosomes extended drug release up to 24 h and showed higher IOP-lowering activity than marketed drug.	2019, [55]
	Span 60	Cholesterol	Dorzolamide HCl	TFH	EE% of remote loaded niosomes was higher (>31%) than passive loaded niosomes, and prolonged drug release more than 8 h was observed.	2017, [56]
	Span 60	Cholesterol	Atenolol	TFH	Maximum EE% between formulated niosomes was 80.7%. When inside the in situ gel, niosomes prolonged drug release and IOP-lowering activity more than 8 h.	2014, [57]
	Span 60	Cholesterol	Brimonidine tartrate	TFH	EE% of the niosomes were in >32, and niosomes showed IOP-lowering activity 3 h longer than commercial drug.	2010, [58]
	Span 60	Cholesterol Chitosan	Timolol maleate	REV	EE% of niosomes was 24.3%, and niosomes extended drug release more than 10 h and IOP-lowering activity up to 8 h with peak at 3 h.	2009, [59]
	Span 60	Cholesterol Carbopol	Acetazolamide	REV	EE% of prepared niosomes was 43.75%, and carbopol coated niosomes showed more sustained and higher IOP-lowering activity than marketed drug.	2007, [60]
	Span 40, 60	Cholesterol	Acetazolamide	TFH, REV	Span60:chol (7:6) MLV niosomes are the highest entrapment efficiency (>32%), and these niosomes prolonged IOP-lowering activity compared to Span60:Chol (7:4) MLV niosomes.	2005, [36]
	Span 60	Cholesterol Chitosan Carbopol	Timolol maleate	REV	Chitosan coated niosomes formulation showed prolonged decrease in IOP compared to carbopol coated niosomes	2005, [61]
Conjunctivitis	Span 60	Cholesterol	Azithromycin-β-CD	SI, TFH, Hand shaking	EE% of prepared niosomes (>30%) and niosomal in situ gel formulations (>63%) were high, and niosomes prolonged drug release up to 12 h with increased corneal permeation.	2017, [62]
	Span 20, 60, 80 Tween 40, 60, 80	Cholesterol	Lomefloxacin HCI	TFH	EE% of the tween niosomes (>41%) and drug release rate were higher than spans niosomes (>40%), and niosomal formulation healed the infected eyes better than marketed drug.	2017, [63]
	Span 60	Cholesterol	Chloramphenicol	SI	EE% of niosomal formulations were >83%, and niosomes prolonged drug release up to 10 h. Niosomes cured disease with less administration than marketed eye drop.	2012, [64]
Fungal keratitis	Span 60	Cholesterol Dicetyl phospate N-Trimethyl chitosan	Natamycin	TFH	Both uncoated and TMC coated niosomes extended drug release up to 12 h., and TMC coated niosomes are higher mucoadhesion and corneal permeability than uncoated niosomes.	2019, [65]
	Span 60	Cholesterol Chitosan	Fluconazole-hydroxypropyl-β-CD	TFH	EE% of the chitosan-coated niosomes were 61.7%, and chitosan coated niosomes prolonged drug release up to 24 h.	2019, [66]
	Span 60	Cholesterol	Natamycin	TFH	EE% of niosome formulations were high (>65%), and niosomes extended drug release up to 24 h with increased corneal permeation.	2017, [67]
	Span 40, 60	Cholesterol Pluronic L64 Pluronic F127	Voriconazole	TFH	EE% of the prepared formulations was >49%, also niosomes and niosomal in situ gel formulations prolonged drug release up to 8 h.	2016, [68]
	Span 60, 80	Cholesterol	Fluconazole	TFH	Span60:chol (2:1) niosomes had the highest EE% (>84%) and better release kinetics. Niosome incorporated into poloxamer gel showed better antifungal activity than niosomes incorporated into chitosan gel.	2016, [69]
	Span 60	Cholesterol Bile salts Edge activator	Tercanazole	SI	Increased edge activator concentration was decreased EE% of niosomes, and selected formulation had EE% of 95.47%. Niosomes increased corneal permeation of drug.	2016, [70]
	Span 40, 60, 80	Edge activators Cholesterol	Fluconazole	SI	In the prepared formulations highest EE% was 65.73%, and corneal permeation of niosomes were higher than marketed drug.	2012, [71]
Genetic retinal disorders	Tween 80 Poloxamer 188	Chloroquine (CQ) Cationic lipid 2,3-di(tetra decyloxy)propan-1-amine	pCMS-EGFP plasmid	REV	Transfection efficiency of pristine nioplexes was higher than CQ-nioplexes but efficiency of both nioplexes were lower compared to Lipofectamine 2000, and CQ-nioplexes transfected cells in inner layers of retina.	2019, [72]
	Tween 20, 80, 85	Cationic lipid DOTMA Squalene	pCMS-EGFP reporter plasmid	REV	Cell viability was higher for nioplexes transfected cells, but transfection efficiency was lower compared to Lipofectamine 2000, and Tween 20 nioplexes was the highest transfection efficiency (>24.5%) for ARPE-19 cells	2018, [38]
	Tween 60	Lycopene Cationic lipid DOTMA	pCMS-EGFP plasmid	REV	Lycopene was increased transfection efficiency of nioplexes, but the efficiency was lower than lipofectamine 2000. Nioplexes were transfected cells in inner layers of retina.	2017, [73]
	Tween 80	Cationic lipid 2,3-di(tetradecyloxy)propan-1-amine Squalene	pCMS-EGFP plasmid	REV	Cell viability was higher for nioplexes compared to Lipofectamine 2000. in vivo transfection was depending administration route; subretinal injection transfected RPE layer cells but intravitreal injection transfected cells in inner layers of retina.	2014, [74]
Corneal graft rejection	Span 60 Tween 80	Hyaluronic acid	Cyclosporine A	SI	HA-coated elastic niosomes had EE% of >92%, and HA-coated formulations were higher corneal permeability than drug emulsion and Span60 niosomes.	2019, [75]
	Poloxamer 188 Soybean phophadithylcoline	Cholesterol Hyaluronic acid	Tacrolimus (FK506)	Proniosome	HA improved the mucoadhesion niosomes, prolonged the residency of drug, and decreased clearance rate of drug in aqueous humor.	2016, [54]
	Poloxamer 188 Lecithin	Cholesterol	Tacrolimus (FK506)	Proniosome	Drug loaded with EE% of 95.32%, and niosomes enhanced the precorneal permeation with prolonged corneal graft survival	2014, [76]
Ocular infections	Span 60 Tween 40	Cholesterol	Vancomycin	TFH	EE% of niosomes was >46%, and niosomes prolonged drug release more than 24 h with increased antibacterial activity.	2019, [77]
	Span 20, 60, 80	Cholesterol	Lomefloxacin HCI	TFH	Drug EE% of niosomes were >78.1%, and niosomes prolonged drug release up to 8 h with higher antimicrobial activity than free drug.	2017, [78]
	Span 60	Cholesterol Chitosan	Gatifloxacin	SI	EE% of chitosan coated niosomes was >68.9%, and chitosan coating was further improved corneal permeability of niosomes	2015, [79]
	Tween 60, 80 Brij 35	Cholesterol Dicetyl phospate	Gentamicin sulfate	TFH	Drug EE% of prepared niosomes was changed in wide range, but all of the niosomes prolonged drug release up to 8 h.	2008, [80]
Ocular surface diseases	Span 20, 60, 80 Tween 60	Cholesterol	Doxycycline hyclate	TFH with extrusion, REV	Span60 niosomes had the highest EE% (>50%), and niosomes extended release of drug more than 20 h.	2019, [81]
Ocular inflammation	Span 60	Cholesterol	Flurbiprofen	TFH	Niosomal in situ gel formulation extended drug release more than 7 h and maintained drug concentration in the aqueous humor up to 12 h.	2016, [82]
	Span 60	Cholesterol Ethanol	Prednisolone	TFH, SI	EE% of niosomes prepared by TFH was higher than SI, and niosomes extended drug release up to 8 h. Also, niosomes decreased the inflammation more than drug solution.	2014, [83]
Diabetic keratopathy	Span 60 Solulan C24 Sodium cholate	Cholesterol Dicetyl phospate	Naltrexone	REV	All of the niosomal formulations showed no irritation and have good corneal tolerability.	2012, [84]
	Span 60 Solulan C24 Sodium cholate	DCP Stearyl amine Cholesterol	Naltrexone	TFH, REV, DRV, Freeze/thaw	EE% of niosomes prepared from different methods compared, and *Rev.* niosomes showed highest EE%, and niosomes prolonged drug release up to 12 h.	2011, [42]

Abbreviations: Thin-film hydration method (TFH); reverse phase evaporation method (REV); dehydration–rehydration vesicles method (DRV); solvent injection method (SI); dicetylphosphate (DCP); cyclodextrin (CD); hyaluronic acid (HA); voriconazole (VCZ); N-Trimethyl chitosan (TMC); chloroquine (CQ); encapsulation efficiency (EE%); brimonidine tartrate (BRT); intraocular pressure (IOP); retinal pigment epithelium (RPE); cholesterol (chol).

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
