# Peer review of "Niosomal Drug Delivery Systems for Ocular Disease—Recent Advances and Future Prospects"

_nanomaterials, 2020, doi:10.3390/nano10061191_

Round 1

Reviewer 1 Report

The article “Niosomal drug delivery systems for ocular disease; recent advances and future prospects” by Saliha Durak and colleagues reviews the recent literature about noisome-based drug delivery for the treatment of glaucoma, conjunctivitis, and retinal diseases. The manuscript is quite well-organized but, in my opinion, it should be rebalanced.

Most of the review is about the preparation methods and characterization of niosomes (pages 2-17) and only the last section (pages 18-19) is about their application as therapeutics for ocular diseases.

Furthermore, many concepts in section 3 are quite general and basic and, in my opinion, not suitable to a journal devoted to nanoformulations and whose audience is represented by researchers specialized in the field of nanotechnology.

Also, more emphasis should be given to the drug-niosome system instead of extrapolating generalized findings based only on the noisome characteristics (i.e. in discussing entrapment efficiency and in vitro release).

On the other hand, section 4 could be enriched with additional references about the recent exploitation of niosomes for treating ocular diseases. Similarly, in the conclusions, the authors should discuss more in detail current problems in their application for ocular delivery (not just the advantages) and future prospects.

Reviewer 2 Report

This manuscript reviews niosomes as drug delivery agents for ocular diseases. In general the manuscript is well-written, with suitable sections for the synthesis, characterization and uses of the niosomes. Some minor mistakes in English can be taken care of by the authors. Also, I would say, that the Table 2 is helpful, but I wonder if it is a little bit too big. Some ‘crucial points’ of some examples from the literature in this text, could be transferred to the main text, couldn’t they? I would say that you can try to ‘maximize’ the amount of these things in your text, that is, of the points when you make ‘critical comparison’ of the niosomes with other agents, mentioning their advantages, possible limitations and so on. Below you can see some more comments:

Lines 69-73: You can change a little bit this part, there is no need to have big similarity with the corresponding phrases in your Abstract.

Line 111: Optimal compliance in what exactly? In the way that the injections are done?

Line 120: ‘These nanoparticles’… I think you need to mention already something about the composition of niosomes, there. What are they composed of? They are made from the assembly of non-ionic surfactants, you mention above, but you can give some more hints, become a bit more specific already in that point. You describe it later, I know, but you can just give some hint.

Line 231: Check if the word ‘basically’ is OK or if it is rather ‘colloquial’ for a manuscript.

Figure 1: Its resolution, especially in the letters, does not seem so clear. Improve it if possible.

References: Shall the corresponding authors cite some of their papers related to this Review? Sometimes we see that corresponding authors do an excessive amount of self-citations. But now, we do not see any citation by the authors of this manuscript. They have some expertise in the topic, don’t they?

Round 2

Reviewer 1 Report

The revised version of the manuscript is significantly improved and, in my opinion, it now deserves publication in Nanomaterials